# Single-cell analysis of inhibitory efferent neurons of the zebrafish lateral line

**Remy Manuel** ⓘ*, **Aikeremu Ahemaiti, Melek Umay Tuz-Sasik** ⓘ, **Henrik Boije***

Department of Immunology, Genetics and Pathology, Uppsala University, Uppsala, Sweden

* remym@me.com (RM), henrik.boije@igp.uu.se (HB)

## Abstract

The zebrafish lateral line system is a sensory network made up of neuromasts, which contain hair cells to detect water flow. Neuromasts signal via sensory afferent neurons and their activity is modulated by efferent neurons. Inhibitory efferent neurons consist of REN, ROLE, and RELL cells and previous work has shown that neuromasts can be innervated by multiple efferent neurons, suggesting potential functional differences. To explore this, we performed single-cell RNA sequencing on REN, ROLE, and RELL neurons in 5-day-old zebrafish larvae. GO analysis across differentially expressed genes did not reveal pathways that suggest differences in cellular function. Comparing markers for neurotransmitter phenotype showed all inhibitory efferent neurons to be cholinergic, but also expressed genes related to other neurotransmitters. Expression of selected genes related to rhombomere location, axon guidance, or gap junctions was similar across efferent neurons. Expression of genes encoding proteins related to membrane potential suggest that REN neurons might be more sensitive to glutamate and may have different action potential dynamics, although functional validation remains to be done. In addition, we assessed neuromast innervation by ROLE and RELL neurons. We found that both ROLE and RELL neurons synapse to approximately 50% of hair cells within a neuromast, compared to approximately 75% innervation by all inhibitory efferent neurons combined. In addition, we did not observe flow polarity bias by innervating efferent axons. However, we did find that RELL neurons had a lower number of synaptic boutons compared to ROLE, which may reflect differences in synaptic output capacity. Taken that our transcriptional analysis did not reveal major intrinsic molecular differences, but we did observe differences in neuromast innervation, raises the possibility that functional differences, if present, may come from upstream inputs. Future work, such as retrograde tracing, could help map these input partners and clarify how different types of efferent neurons contribute to sensory modulation.

**Data availability statement:** All relevant data for this study are publicly available from the European Nucleotide Archive (ENA) repository (https://www.ebi.ac.uk/ena/browser/view/PRJEB110328).

**Funding:** Financial support came from: the Kjell and Marta Beijers Foundation; the Jeanssons Foundation; the Carl Tryggers Foundation; the Swedish Brain Foundation; the Swedish Research Council; the Magnus Bergvalls Foundation, the Royal Swedish Academy of Sciences; the Ake Wibergs Foundation; Olle Engkvist Stiftelse; the Ragnar Söderberg Foundation, and the Swedish Foundation for Strategic Research. The SNP&SEQ Platform is also supported by the Swedish Research Council and the Knut and Alice Wallenberg Foundation. The funders had no role in study design, data collection and analysis, decision to publish, or preparation of the manuscript.

**Competing interests:** The authors have declared that no competing interests exist.

## Introduction

The zebrafish lateral line is made up by sensory organs called neuromasts. These neuromasts consist of hair cells, which detect the surrounding flow of water, crucial information for behaviours such as rheotaxis, predator avoidance, and schooling [1,2]. The zebrafish lateral line is frequently used as a model system to understand the formation and function of sensory circuits. Notably, the hair cells in neuromasts are similar to those in the mammalian ear and several zebrafish models exist to study hearing disorders [3] or screen drugs and drug-like compounds protecting against hair cell death [4,5].

Neuromasts are innervated by afferent and efferent neurons to detect and modulate hair cell activity. In zebrafish, sensory afferent neurons are clustered in two main ganglia situated on either side of the ear: the anterior lateral line (ALL) ganglion and the posterior lateral line (PLL) ganglion [6]. The efferent neurons are positioned in the brain: the diencephalic efferent of the lateral line (DELL) are located in the diencephalon, and the rostral efferent nucleus (REN) and caudal efferent nucleus (CEN) are found in the rhombencephalon [7]. Within the CEN, we can identify two types of neurons based on their axon projection path: the rhombencephalic octavolateral efferent neuron (ROLE) and the rhombencephalic efferent neuron to the lateral line (RELL). DELL neurons are dopaminergic and provide excitatory input [8,9], while REN and CEN neurons are cholinergic and attenuate lateral line sensitivity [10].

Previously, we assessed how the three morphologically distinct efferent neurons, REN, ROLE, and RELL innervate the neuromasts of the zebrafish lateral line [11]. We found that a single neuromast could be innervated by more than one inhibitory efferent neuron. This dual innervation by REN, ROLE and RELL cells may hint towards functional differences. For instance, efferent neurons may provide orientation selective inhibition, similar to that of sensory afferents, ensuring direction sensitivity [12]. Alternatively, efferent neurons may receive input from different sources depending on behaviours. This would allow for modulatory control during feedback events [13,14] or feedforward inhibition [15,16]. In addition to differences in connectome, differences in excitability may exist, providing a control over the power by which inhibitory efferent neurons silence neuromasts.

Identifying whether functional differences between inhibitory efferent neurons exist could provide new insights into the biology of sensory modulation and associated diseases. To that end, we compared gene expression profiles between individual REN, ROLE, and RELL neurons, with the aim to find differentially expressed genes (DEGs) that could contribute to differences in function. We used 5-day-old transgenic zebrafish larvae with mosaic labelling of inhibitory efferent neurons and collected individual cells for single-cell RNA sequencing. Although we did identify DEGs, the biological processes these genes are involved in does not directly suggest functional differences. Next, we performed more direct analysis of gene expression, where the main purpose was to reveal gene-on/gene-off differences between the inhibitory efferent neuronal types. Based on their gene expression, all three efferent neurons have the same cholinergic neurotransmitter phenotype, but also express many genes related to other neurotransmitter pathways. Next, we compared expression of genes

related to rhombomere position, axon guidance, gap junctions, and membrane potential. Overall, the genes analysed showed similar expression patterns among the three efferent neuronal types, with the exception of subtle differences in genes related to membrane potential, suggesting an increased glutamate sensitivity in REN neurons. Lastly, we compared the innervation of neuromasts between ROLE and RELL neurons and observed that the number of synaptic boutons was lower for RELL neurons, but the number of hair cells innervated was equal. In addition, innervation of hair cells did not correlate with hair cell flow polarity.

Although we cannot rule out intrinsic differences among efferent neurons, our gene expression analysis did not reveal strong transcriptional divergence. Instead, we believe it more likely that functional differences are derived at the level of upstream input. The observed difference in synaptic boutons between ROLE and RELL in neuromasts supports this, as it hints to differences in input capacity between these efferent cell types. Future studies, where individual efferent neurons are activated through cell patching could test this hypothesis. In addition, input partners of the inhibitory efferent neurons could be revealed using retrograde viral tracing, further uncovering variability in sensory modulation by inhibitory neurons.

## Results and discussion

### Individual inhibitory efferent neurons

To reliably collect individual efferent neurons, we used Tg(UAS:Tomato), which mosaically labels neurons, to mark single efferent cells [11]. At 4 days post fertilisation (dpf), larvae were screened using a confocal microscope and we recorded the inhibitory efferent subtype: REN, ROLE or RELL, or as non-projecting (NP) if it did not send an axon to the lateral line (Fig 1A-C). At 5 dpf, neurons were collected and processed for RNA amplification. In addition, we included four samples where we collected the entire rhombomere region including the efferent neurons (hindbrain: HB).

### Sequencing and neuron/glia genes

We assessed the number of reads (nReads) between REN (51,528,603), ROLE (48,856,211), RELL (40,274,182), NP (39,153,029), and HB (46,607,819) and found no significant differences (K = 0.9907, p = 0.91, Fig 2A). These reads revealed a total of 25,410 unique gene transcripts (nGenes) in our dataset, with HB (17,617) showing a higher number of nGenes compared to REN (7,592), ROLE (6,276), RELL (7,090), and NP (5,940) groups (K = 13.48, p = 0.0092, Fig 2B), which is to be expected given the heterogeneity of the bulk sample. We also assessed if samples with higher nReads had a higher number of nGenes, but found this to be not the case (r(54)=−0.031,p = 0.82).

To verify the identity of our samples, we checked for the presence of neuron specific genes (*elavl3* and *snap25*) [17,18] as well as genes commonly expressed in glia (*nes*, *her4.3*, *s100b*, and *gfap*) [19]. In addition, we looked for the presence of genes known to be expressed in the lateral line inhibitory efferent neurons, such as *isl1* [20,21] and *dmrt3a* [11]. We found robust expression for neuronal and efferent neuron markers, while glia markers were weakly expressed (Fig 2C).

We also assessed the neuron *versus* glia gene expression ratios in our samples, as inadvertent co-collection of off-target mRNA from glia is not uncommon in patch-seq approaches [22]. Five neurons had no expression of glia markers (1x ROLE, 1x RELL, 3x NP). We found that only one cell had a ratio below 1.0 (1x RELL = 0.80), suggesting a stronger expression of glia markers compared to neuron marker genes. This cell was not excluded in our downstream analysis. All other samples showed ratios above 1.0 (1.40 to 59.88; Fig 2D). These findings indicate that we collected our neurons of interest and that the contamination by glia was kept to a minimal.

### Differentially expressed genes

Due to low RNA concentration, we excluded 8 samples from our analyses (4x REN, 2x ROLE, 1x RELL, 1x NP), leaving us with 16x REN, 12x ROLE, 11x RELL, 11x NP, and 4x HB samples. For our downstream analysis, we kept cells/samples

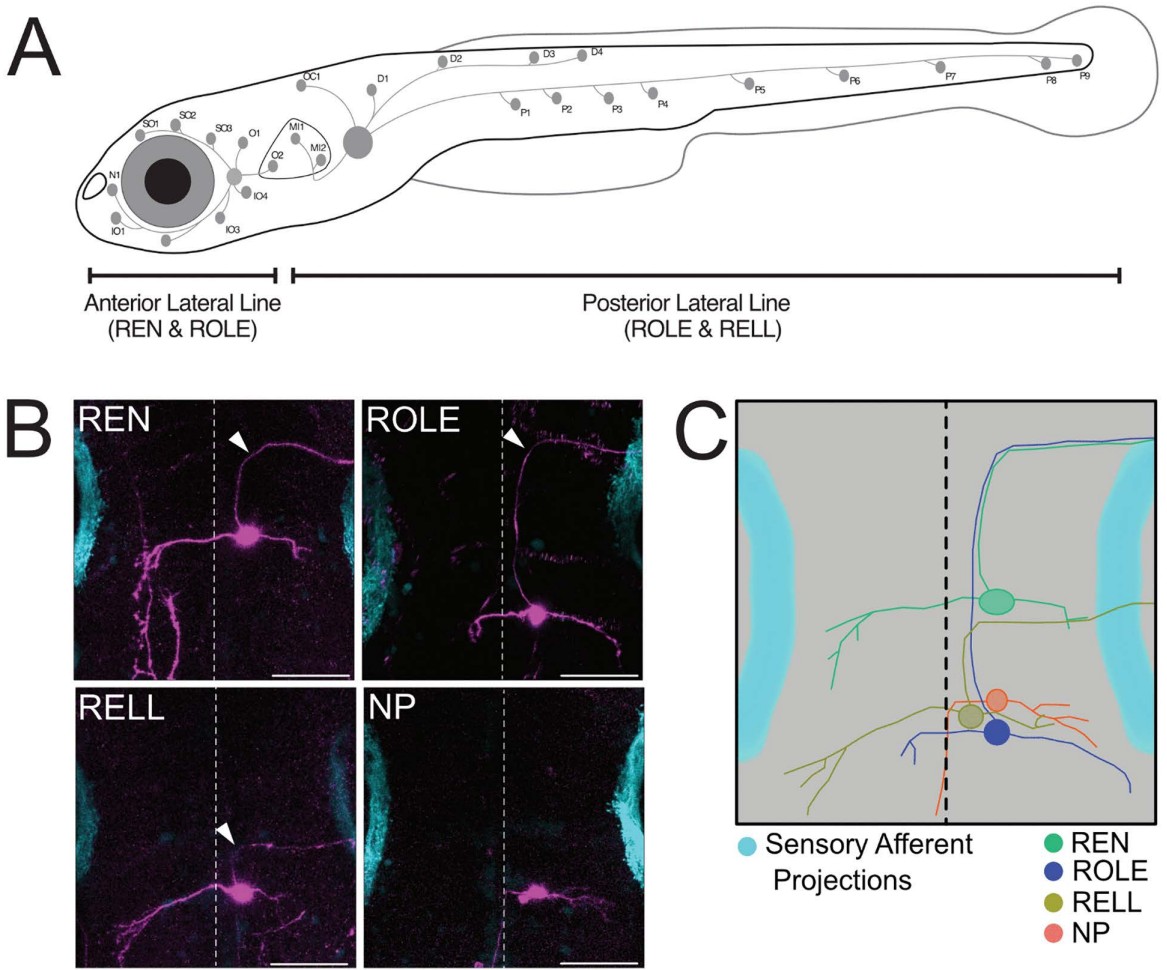

**Fig 1. Morphology of REN, ROLE and RELL neurons. (A)** Schematic view of the projection paths and neuromast locations of the zebrafish Anterior Lateral Line and Posterior Lateral Line at 5 dpf (adapted from Manuel and colleagues [11]). **(B)** Top view of the rhombomere region of 5 dpf old Tg(*dm-rt3a*:GAL4;UAS:Tomato) larvae, where a single inhibitory efferent neuron (REN, ROLE, RELL) or single neurons that do Not Project to the lateral line (NP) can be seen (magenta). Lateral line sensory afferent projections (cyan) marked by Tg(Hgn39D) were included for orientation and determining efferent neuron positions. White arrows indicate axon projection towards the lateral line. **(C)** Schematic overlay of REN, ROLE, RELL and NP neurons showing their position in relation to each other. Dashed lines represent the midline. Scale bar = 50 μm.

that had at least 100 detected genes and required that these genes were expressed in at least 3 cells/samples. No samples were excluded based on this restriction.

We first identified the top 2000 most variable genes (variable genes were identified based on how their expression changed between samples) to calculate the average expression and dispersion for each gene, and these genes were then placed into bins. For dispersion within each bin, a z-score was calculated. Through this approach, the relationship between average expression and variability was controlled. After dimensionality reduction, we saw a clear separation of the HB samples, but also that two REN samples stood out (Fig 3A). Subsequent MultiQC report did not reveal any significant bias for these samples and the top 20 expressed genes had more or less the same expression level similar to the other samples. These samples were therefore not excluded for the downstream analysis.

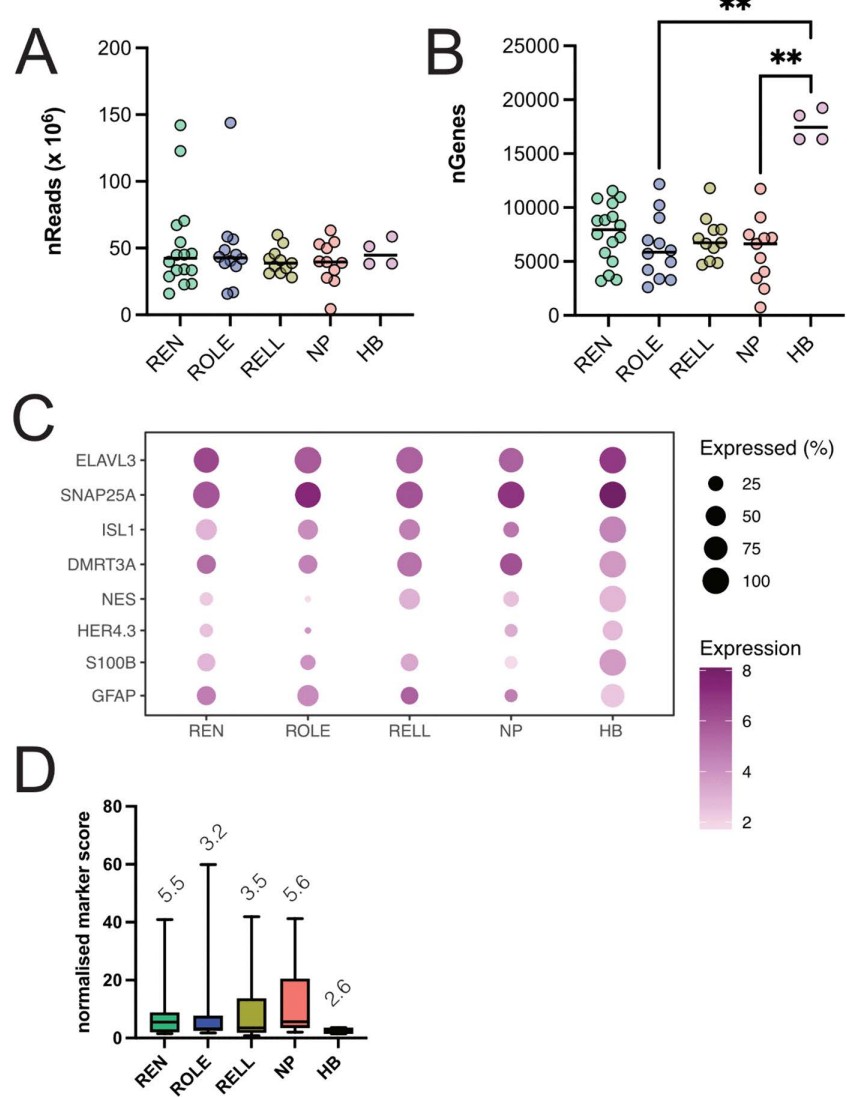

**Fig 2. Expression of genes related to neurons and glia. (A)** Number of reads for each sample. Line represents the mean for each group. **(B)** Number of detected genes in each sample. Line represents the mean for each group. **(C)** Dot-plot of genes related to neurons, inhibitory efferent neurons, and glia. **(D)** Box and whiskers plot (median with 2.5-97-5 percentile) of the neuron/glia expression ratios for REN, ROLE, RELL NP, and HB. Numbers above the plots indicate median values per cell group, higher ratios indicate stronger neuron marker expression compared to glia marker expression. $p = 0.01$ (**).

Next, we proceeded with identifying DEGs between the REN, ROLE, RELL and NP samples. First, we compared NP *vs.* REN, ROLE, RELL, and found 34 differentially expressed genes (Fig 3B). Next, we plotted the DEGs among the REN, ROLE and RELL neurons. We compared REN *vs.* ROLE and RELL, to identify potential REN and CEN related gene expression, and found 17 DEGs (Fig 3C). We also compared REN *vs.* ROLE (44 DEGs), REN *vs.* RELL (15 DEGs), and ROLE *vs.* RELL (49 DEGs) to pinpoint unique genes per subtype (Fig 3D-E). Although we were able to identify a good number of DEGs between the NP and the inhibitory efferent neurons, as well as between REN, ROLE and RELL cells, a GO analysis for biological processes (ShinyGP 0.85.1; https://bioinformatics.sdstate.edu/go/) did not indicate that these DEGs would result in functional differences (see Legend of Fig 3).

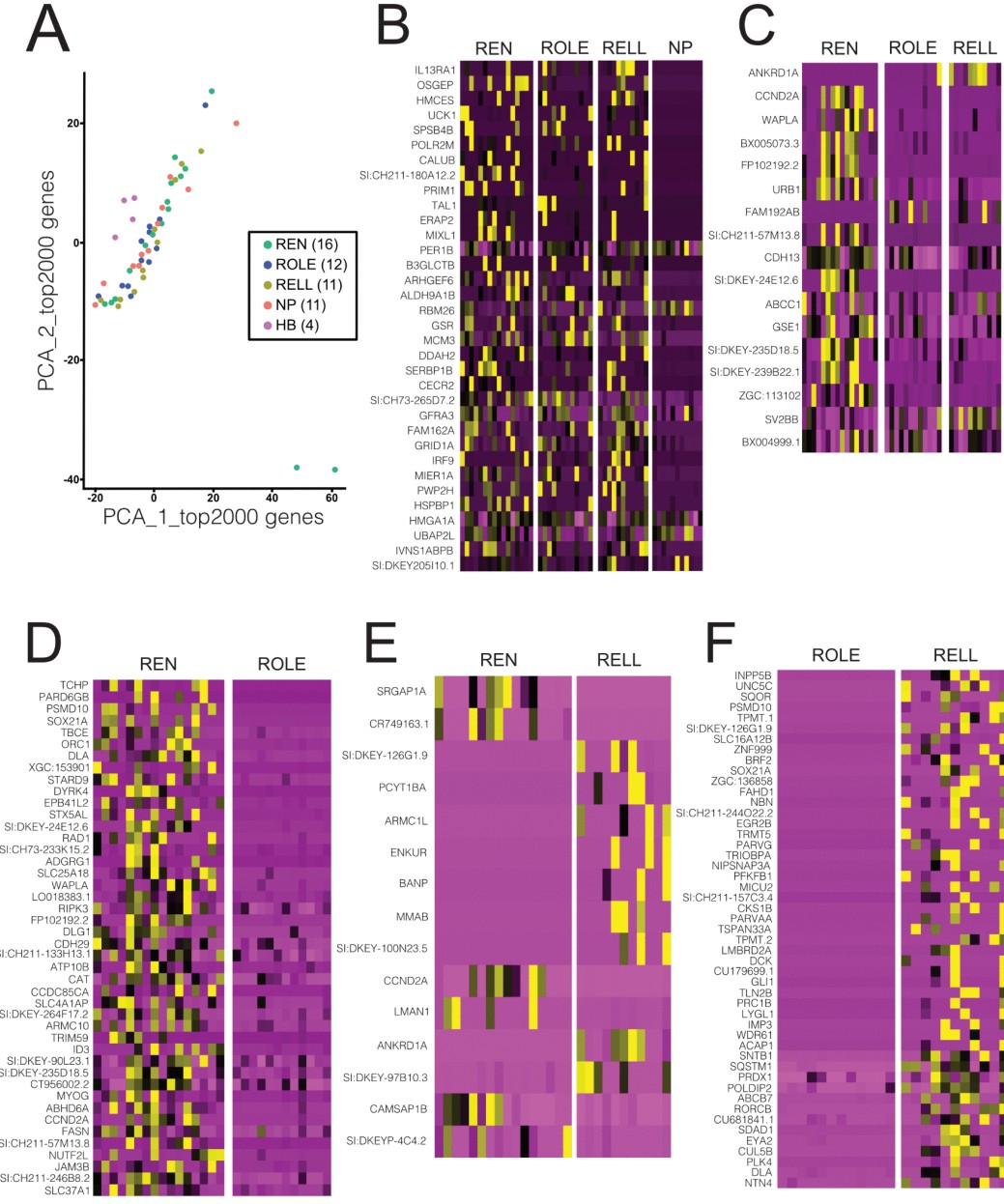

**Fig 3. Differentially expressed genes lateral line inhibitory efferent neurons. (A)** PCA representation based on the top 2000 expressed genes in the cells retained in the dataset. HB samples clustered separate from the other samples. Two REN were plotted as outliers (bottom right). **(B)** Heatmap of differentially expressed genes when comparing NP *vs.* REN, ROLE, RELL. **(C)** Heatmap of differentially expressed genes comparing REN *vs.* ROLE, RELL. **(D-F)** Heatmaps of differentially expressed genes comparing: REN *vs.* ROLE **(D)**, REN *vs.* RELL **(E)**, or ROLE *vs.* RELL **(F)**. Go Analysis results for the DEGs: *B*: none; *C*: 14 processes, including tripeptide transport, oligopeptide transport, maturation of lsu-rRNA and 5.8s rRNA (nGenes 1 each); *D*: 15 processes, including myelination periphery axons, Schwann cell development, lateral line glia cell development, oligodendrocyte development (nGenes 2 each) and 1 process involves Membrane organization (nGenes 4); *E*: none; *F*: substrate adhesion-dependent cell spreading (nGenes 4) and cell substrate adhesion (nGenes 4)..

## Neurotransmitter genes

To characterise the neurotransmitter phenotype of the inhibitory efferent neurons, we plotted the expression of inhibitory and excitatory neurotransmitter marker genes (Fig 4A). None of the efferent neurons showed expression of genes associated with an excitatory phenotype: vGluT1 (*slc17a7b*), vGluT3 (*slc17a8*) or vMAT2 (*slc18a2*). We did find expression for

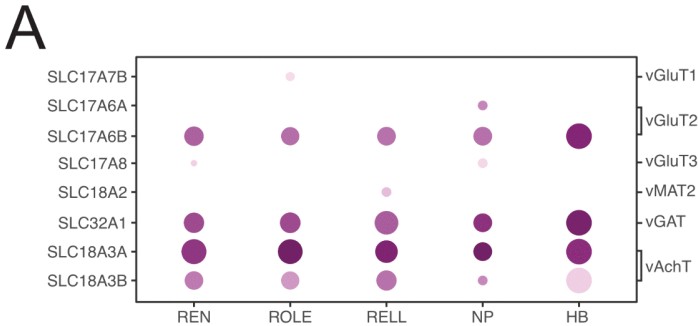

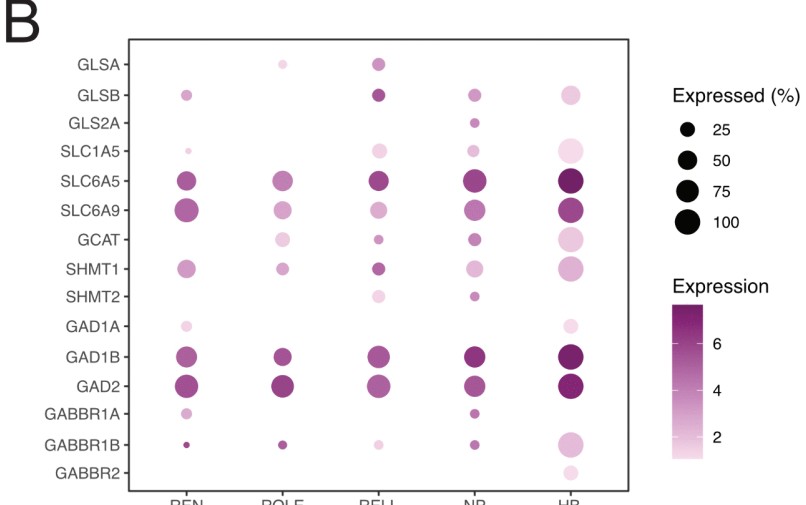

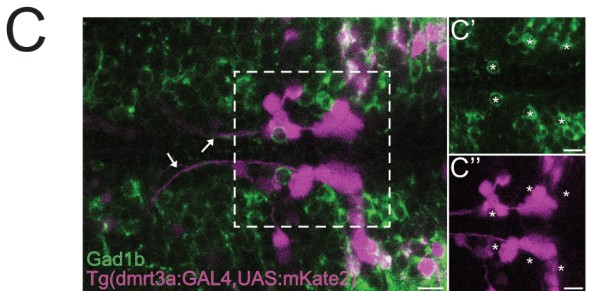

**Fig 4. Expression of neurotransmitters and their signalling pathways. (A)** Dot-plot of genes related neurotransmitter phenotype. **(B)** Dot-plot of genes related to glutamate, glycine, and GABA signalling. Note that *gls2b* (<1.0 scaled expression) is not plotted. **(C)** immunoreactivity for Gad1b antibody (Green; **C'**) in Tg(*dmrt3a*:GAL4,UAS:mKate2) marking neurons in the REN and CEN, including the REN, ROLE, and RELL inhibitory efferent neurons (magenta. **C"**). Arrows indicate efferent axons projecting towards the lateral line, asterisks mark a selection of Gad1b-positive neurons near mKate2-postive REN and CEN neurons. Scale bar = 10 μm.

*slc17a6b,* but not for *slc17a6a,* both encode for vGluT2. In contrast, REN, ROLE and RELL displayed robust expression of genes for vGaT (*slc32a1*) and vAchT (*slc18a3a* and *slc18a3b*), with vAchT being the most prominent. The expression for vAchT genes was expected, as studies have shown that inhibitory efferent neurons act via acetylcholine through the cholinergic a9 receptor found in neuromast [23–25]. The presence of *slc17a6b* (vGluT2) and *slc32a1* (vGaT) expression in addition to *slc18a3a* and *slc18a3b (*vAchT) may be indicative of a multi-neurotransmitter phenotype [26]. To explore this further, we looked for other genes associated with glutamate (vGluT2), and glycine or GABA (vGAT) signalling.

**Glutamate.** It is not uncommon for neurons to express multiple neurotransmitters. For instance, glutamatergic and GABAergic cells can express genes associated with cholinergic signalling, but fail to accumulate cholinergic proteins [27]. The reverse may be possible in our neurons, where expression of glutamatergic genes may not result to proteins.

Although we observed expression for vGluT2, our efferent neurons lack robust expression of genes for glutaminase (*glsa* and *glsb)* and glutaminase 2 (*gls2a* and *gls2b* [Fig 4B]), the enzymes responsible for producing glutamate from glutamine [28]. In addition, efferent neurons did not express Nnat1 (*slc38a1* not found in the dataset), a transporter that facilitates the uptake of glutamine in glutamatergic neurons [29], and only a handful of cells displayed low levels of the neutral amino acid transporter Asct2 (*slc1a5* [Fig 4B]). In addition, gene expression for high-affinity excitatory amino acid transporters (*eaat*), which regulate extracellular glutamate levels [30], was not found in the dataset. Combined, these results make it unlikely that the efferent neurons are glutamatergic.

**Glycine.** Both GlyT2 (*slc6a5*) and GlyT1 (*slc6a9*) were highly expressed in our samples (Fig 4B). GlyT2 enables the (re-)uptake of glycine from the synaptic cleft. Glycine is a coactivator of NMDA receptors for glutamate [31] and neuromast hair cells signal via glutamate release to sensory afferent neurons [32]. If translated into functional proteins, glycine transporter expression could influence glycine availability at the level of afferent synapses. Here, the inhibition would not be an immediate response to neuromast activity, but rather attenuate signalling during resting states or prevent over-stimulation during long-term activation of neuromasts (*e.g.*, constant flow exposure). GlyT1 is found on the post-synaptic membrane of glycinergic synaptic connections [31], suggesting that efferent neurons receive glycinergic input. Both glycine c-acetyltransferase (*gcat*) and hydroxumethyltransferase (*shmt2*) regulate glycine production in mitochondria [33,34]. Expression levels of *gcat* are relatively low and seen in only a few neurons (Fig 4B). Similarly, we did not detect robust expression of *shmt1* or *shmt2* (Fig 4B). Combined, these results suggests that efferent neurons do not signal via glycine.

**GABA.** Efferent neurons showed robust expression of *gad2* and *gad1b* (but not *gad1a* [Fig 4B]), encoding for enzymes required for the synthesis of GABA [35,36]. In mice, nearly all cholinergic neurons of the forebrain express the cellular machinery necessary to release the inhibitory neurotransmitter GABA [37,38]. The potential GABAergic action of efferent neurons seems not aimed at neuromast cells, as these lack the necessary GABA receptors [25]. In the mammalian auditory system, GABA acts upon the inhibitory efferent neuron itself to modulate its own acetylcholine release [39,40]. However, our inhibitory efferent neurons lack GABA receptor 1a and 2 (*gabbr1a* and *gabbr2*; Fig 4B) through which this self-inhibition operates [40]. Previous studies have shown that GABA reduces afferent activity of the Xenopus lateral line [41] and that GABA acts directly on afferent projections in the mouse auditory system [42]. In addition, *in situ* hybridisation performed on zebrafish has shown the expression of *gabbr1a* in the afferent ganglions of the anterior and posterior lateral line [43,44]. Taken together, these observations are consistent with the notion that GABA signalling may act at the level of sensory afferent neurons in the zebrafish as well.

To validate our finding, we performed immunohistochemistry against Gad1b, but found no immunoreactivity in the inhibitory efferent neurons (Fig 4C). This observation is in line with the stable Tg(*gad1b:GFP)* zebrafish reporter line [45], which does not mark the inhibitory efferent neurons. A discrepancy between gene expression and protein detection could come from asynchronous transcription and translation observed in maturing neurons, where genes are transcribed into mRNA long before corresponding proteins are detected [46]. In addition, we cannot rule out that Tg(*gad1b:*GFP) does not label all GABAergic cells in a given tissue, as incomplete labelling can result from genomic silencing of the integration site [47].

 

Still, although gene expression data suggests efferent neurons may be GABAergic, transgenic labelling and immunohisto-chemistry suggest that they are not.

As we cannot exclude the possibility of contamination by surrounding cells in our collected single cell samples, it is possible that detected transcripts came from other sources. By collecting the entire cell, as opposed to only its nucleus, there is a risk of co-collecting synapses from other neurons connected to our cells of interest. It is known that for many genes, including those involved in neurotransmission, mRNAs are being translated in the synapse [48,49]. Although this should be considered as a potential contamination risk, studies have also shown that the abundance of mRNA in the cell body is much greater than what is found in the axon or synapses [50].

## Rhombomere genes

To maintain strict rhombomere (r) boundaries several Eph-receptors are segmentally expressed in the hindbrain, in a complementary pattern to EphrinBs for which they have high affinity: Epha4 in r3 and 5 is complementary to Ephrinb3 in r2, 4, and 6, while Ephb4 in r2, 3, 5, and 6 is complementary to Ephrinb2 in r1, 4, and 7 [51–53]. However, during early development, REN neurons migrate from r4 to r6, while ROLE and RELL neurons migrate from r5 to r7 [52]. We therefore wondered if we could detect expression of rhombomere boundary genes in the efferent neurons and if these would correspond to their current location within the hindbrain. We found that all three efferent neuronal types strongly expressed *efnb2a*, but almost none expressed *epha4* or *ephb4* (Fig 5A). The lack of rhombomere specific Eph-receptor or EphrinB expression should not come as a surprise. A low level of *epha4* and *ephb4* expression in our neurons is in line with their overall decreased expression during development [54]. In addition, our cells were collected at 5 dpf, a developmental

A

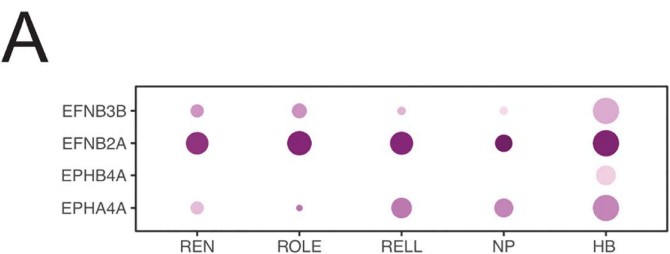

B

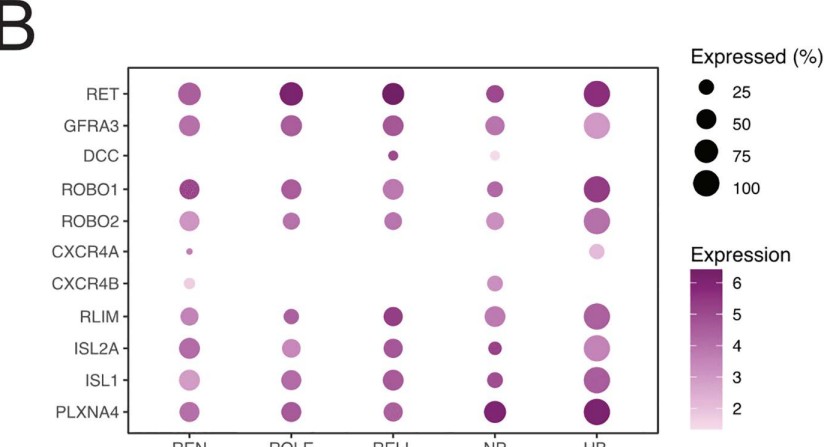

**Fig 5. Expression of genes related to rhombomeres and axon guidance. (A)** Dot-plot of genes related to rhombomere position. **(B)** Dot-plot of genes related to axon guidance and those enabling axons to exit the nervous system.

stage at which the strict rhombomere boundaries are no longer maintained, and thus the need for expression of rhombomere boundary genes is gone.

## Axon guidance genes

The rearranged during transfection (Ret) receptor enables sensory afferent pioneer neurons to follow the primordium of the lateral line, which migrates along the lateral side of larvae and disperses clusters of cells destined to become neuromasts [55]. Here, we found that our inhibitory efferent neurons also expressed this receptor, as well as expression of its co-receptor *gfra3* (Fig 5B). This receptor/co-receptor complex binds Artemin [52], an axon guidance cue produced in neuromasts through the expression of *artnb*, particularly during the maturation of the amplifying support cells (amp-SC) and mantle cells, as well as both the dorsoventral (DV) and anteroposterior (AP) cells [24]: two subpopulations that act as progenitors for new hair cells [56]. Although inhibitory efferent neurons do not connect to these cells [24], their release of Artemin could serve as an attractant. For instance, during early development, where collateral projections from the inhibitory efferent axon can appear hours after the axon's growth cone has passed the neuromast [11], or during stitching events where new neuromasts are formed [57]. Here, sensory afferent neurons innervate the new neuromasts long before hair cells are formed [57] and inhibitory efferent neurons may thus do the same. Efferent axon guidance via Ret-Artemin signalling could also explain why we observed that inhibitory efferent projections still projected towards neuromasts in the absence of hair cells in *atoh1a* crispants [58].

In mice, DCC prevents motor neurons from leaving the central nervous system and the ROBO1 and ROBO2 receptors promote exiting the central nervous system, as they inhibit DCC [59]. Our inhibitory efferent neurons lacked expression of *dcc*, and showed robust expression of both *robo1* and *robo2*, in line with having axons that project to the periphery. Both CXCR4 and CXCR7 have been shown to promote motor neuron exit in mice, but expression of these receptors was not prominent in efferent neurons (*cxcr4a* and *cxcr4b*) or not found in our dataset (*cxcr7* [Fig 5B]). However, the lack of expression of CXC-receptor genes could also mean that, at 5 dpf, we have missed their expression window. Alternatively, it is possible that these receptors do not play a role in axon guidance of efferent neurons.

Rohon Beard neurons have both a central and a peripheral axon branch, each expressing different axon guidance receptors [60]. The branch leaving the central nervous system requires expression of *isl2* and *rlim*. The inhibitory efferent neurons show robust expression of *rlim*, *isl2a* and *isl1*. In addition, the inhibitory efferent neurons express *plxna4* and lack expression of *tag1*, similar to the exiting axon branch of Rohon Beard neurons [60].

The expression of genes associated with axons that exit the central nervous system, and the absence of genes that prevent this, is consistent with molecular features observed in Rohon Beard and motor neurons when it comes to guiding axons out of the central nervous system.

## Gap junction genes

Inhibitory efferent neurons had robust expression of genes for gap junction support proteins, such as *nbeaa*, *nbeab*, and *tjp1b* [61], as well as expression of *gjd1a*, a post-synaptic protein for gap junctions (Fig 6A). In contrast, no expression for *gjd2*, a pre-synaptic gap junction protein [62], was observed. Although there was some expression for *gjd2b* (Fig 6A), it appears that efferent neurons are more likely to receive input via gap junctions, rather than communicate through gap junctions themselves. This notion is supported by a lack of gap junction genes being expressed in cells of the neuromast [25].

Interestingly, the Mauthner cell forms gap junctions by expressing *gjd2a* and is known to form connections with cells that express *gjd1a* [62]. The idea that the Mauthner cell connects to one or more inhibitory efferent neurons has already been proposed [63], but thus far the existence of such connections have not been shown. Using transgenic lines, where fluorescent proteins are fused to gap junction proteins found in the inhibitory efferent neurons, may reveal where gap junctions are made, and potentially reveal post-gap junction partners.

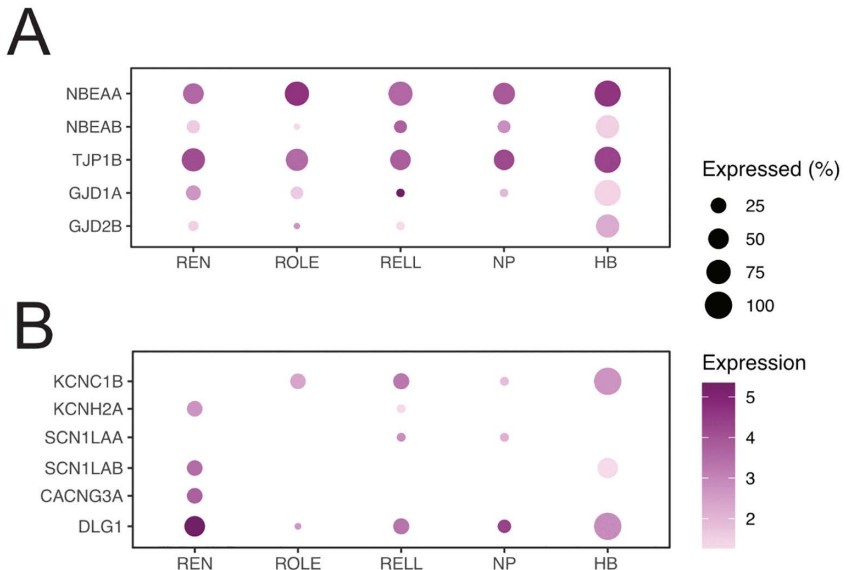

**Fig 6. Expression of genes related to gap-junctions and membrane potential. (A)** Dot-plot of genes related to gap-junctions and their support proteins. Note that *gjd2* (<1.0 scaled expression) is not plotted. **(B)** Dot-plot of selected genes related to glutamate sensitivity and membrane potential.

## Membrane potential genes

We explored if there was differential expression of genes encoding proteins related to membrane potential or transference of action potentials, which could underlie difference in electrophysiological properties. Although the expression of the majority of the genes we looked at was equal among inhibitory efferent neurons, we found that only REN neurons lacked *kcnc1b* expression (Fig 6B). The *kcnc1b* gene encodes for the Kv3.1 protein, which mediates the rapid repolarisation following action potentials and has been proposed as a mechanism to limit the amount of $Ca^{2+}$ entering the cell [64]. The REN neurons did express *kcnh2a*, *scn1lab*, and *cacng3a*, while ROLE and RELL neurons did not (Fig 6B). The *kcnh2* gene encodes multiple transcripts in the ERG family of potassium channels, which serve to stabilise neuronal resting membrane potentials, rather than shaping action potentials [65], and its expression has been linked to a lower density of dendritic spines [66]. The *snc1lab* gene encodes a neuronal voltage-gated sodium channel alpha subunit [67], where knockout animals displayed sudden electrical discharges in the brain and an increased sensitivity to GABA antagonists [68]. Expression of *sncl1aa* could compensate for the absence of *snc1lab* expression [69] as both produce the $Na_v1.1L$ protein. However, the *sncl1aa* gene is not expressed in inhibitory efferent neurons, suggesting that ROLE and RELL cells do not possess $Na_v1.1L$ subunits in their sodium channels. The Tarp-gamma 3 protein (*cacng3a*) is involved in the trafficking of AMPA glutamate receptors to somatodendritic compartments and regulates their gating properties there [70]. For instance, Tarp subtypes determine the kinetics of AMPA receptors and dose-dependently control AMPA receptor gating [71]. In our DEG analysis, we found that the *dlg1* gene was associated with REN neurons (in REN *vs* ROLE, Fig 3D). The DLG-family of proteins has been shown to modulate calcium influx in post synaptic neurons [72] and are involved in the trafficking of AMPA receptors [73]. Although these genes are only a fraction of the total number of genes involved in membrane and action potentials, their importance for shaping electrophysiological properties are well documented. Combined, these differentially expressed genes are consistent with potential differences in excitability or action potential kinetics, but electrophysiological validation is required.

## Neuromast innervation

Although we did observe some DEGs among the different types of inhibitory efferent neurons, their gene expression profiles appeared very similar, suggesting that functional differences are not intrinsic. We therefore explored whether the efferent neurons displayed unique anatomical innervation of neuromasts.

To assess neuromast innervation by individual ROLE or RELL efferent neurons, we again used Tg(*dmrt3a*:GAL4,UAS:Tomato), which labels efferent neurons in a mosaic manner [11], and crossed this to Tg(*myo6b*:hs:eGFP), which labels the neuromast hair cells [58]. Neuron identity was determined through confocal imaging at 5 dpf, and innervation of the P1 and P2 neuromast was assessed at 10 dpf, when the system is more matured. In addition, we assessed innervation by Tg(*dmrt3a*:GAL4,UAS:mKate2), which labels all inhibitory efferent neurons, to assess full innervation (FULL). We focussed on the P1 and the P2 neuromasts as their opposing directional tuning: horizontal flow (P1) or vertical flow (P2), accompanied by distinct afferent innervation, is well-described in literature (Fig 7A) [74,75].

Quantification of the number of hair cells in the analysed neuromasts revealed no significant differences among the ROLE, RELL, and FULL groups (Fig 7B). Next, we set out to compare the number of synaptic boutons by inhibitory efferent projections (Fig 7C). A two-way ANOVA revealed a significant source of variations caused by efferent neuron type ($F_{(2,49)}=5.830$, $p=0.005$). A subsequent ANOVA ($F_{(2,52)}=6.016$, $p=0.005$) revealed that the number of boutons from RELL neurons (8.0±3.7) was significantly lower than those of ROLE (14.4±6.2; $p=0.026$) and FULL (15.4±5.4; $p=0.003$) innervations (Fig 7D). When the data for P1 and P2 neuromast was split and analysed separately, the trend of fewer boutons in RELL neurons remained, but statistical significance was reduced (Fig 7E-F). We also investigated how many of the hair cells in a neuromast were in close proximity of a synaptic boutons and found that ~50% of the hair cells in a single neuromast were close to boutons from a ROLE or RELL efferent (Fig 7G-H). Surprisingly, when we assessed the number of hair cells for FULL innervation, we found that only ~75% of hair cells were in close proximity of efferent boutons (Fig 7G-H). This would suggest that ~25% of hair cells do not receive inhibitory input.

In contrast to our observation, a study using electron microscopy found that all mature hair cells in a neuromast are connected by an efferent synapse [76]. One explanation for the reported differences could be that we used a transgenic zebrafish line which marks cells expressing *myo6b*, a gene also expressed in immature hair cells [77,78]. Although the majority of hair cells should be mature at 10 dpf [79], other studies reported that the percentage of active postsynaptic afferent terminals within a neuromast of a 13–17 dpf zebrafish was found to be between ~50–70% [80,81]. It is thus possible that our quantification includes a portion of immature or inactive hair cells that lack inhibitory input.

In support of our findings, one study found that ~70% of the hair cells in a neuromast are inhibited during events of motor activity [16]. Interestingly, this study also found a correlation between the hair cells being silenced and their flow polarity. Here, all hair cells with a posterior deflection were silenced and only a subset of those with an anterior deflection. To assess if innervation by ROLE and RELL is biased for flow polarity, we oriented our imaged neuromast to align along the anterior-posterior axis of the larvae and mapped the location of the efferent boutons (Fig 7I) as well as the connected hair cells (Fig 7J). Analysing these 'maps' did not reveal a clear bias associated with flow polarity. In fact, innervation patterns showed great randomness in position and number of potentially connected hair cells, not only among REN or RELL cells, but also for the FULL innervation patterns. This seemingly random innervation pattern and lack of an obvious polarity bias may be indicative that sensory modulation occurs at a broader network level, rather than at a single neuromast level.

## Concluding remarks

Our investigation into differential gene expression in the inhibitory efferent neurons of the zebrafish lateral line revealed that REN, ROLE, and RELL neurons have highly similar gene expression profiles: all types exhibit a cholinergic identity and expressing the same genes related to other neurotransmitter systems, axon guidance, and gap junction signalling. While subtle differences were observed in genes related to membrane potential, we believe these to be modest, and not

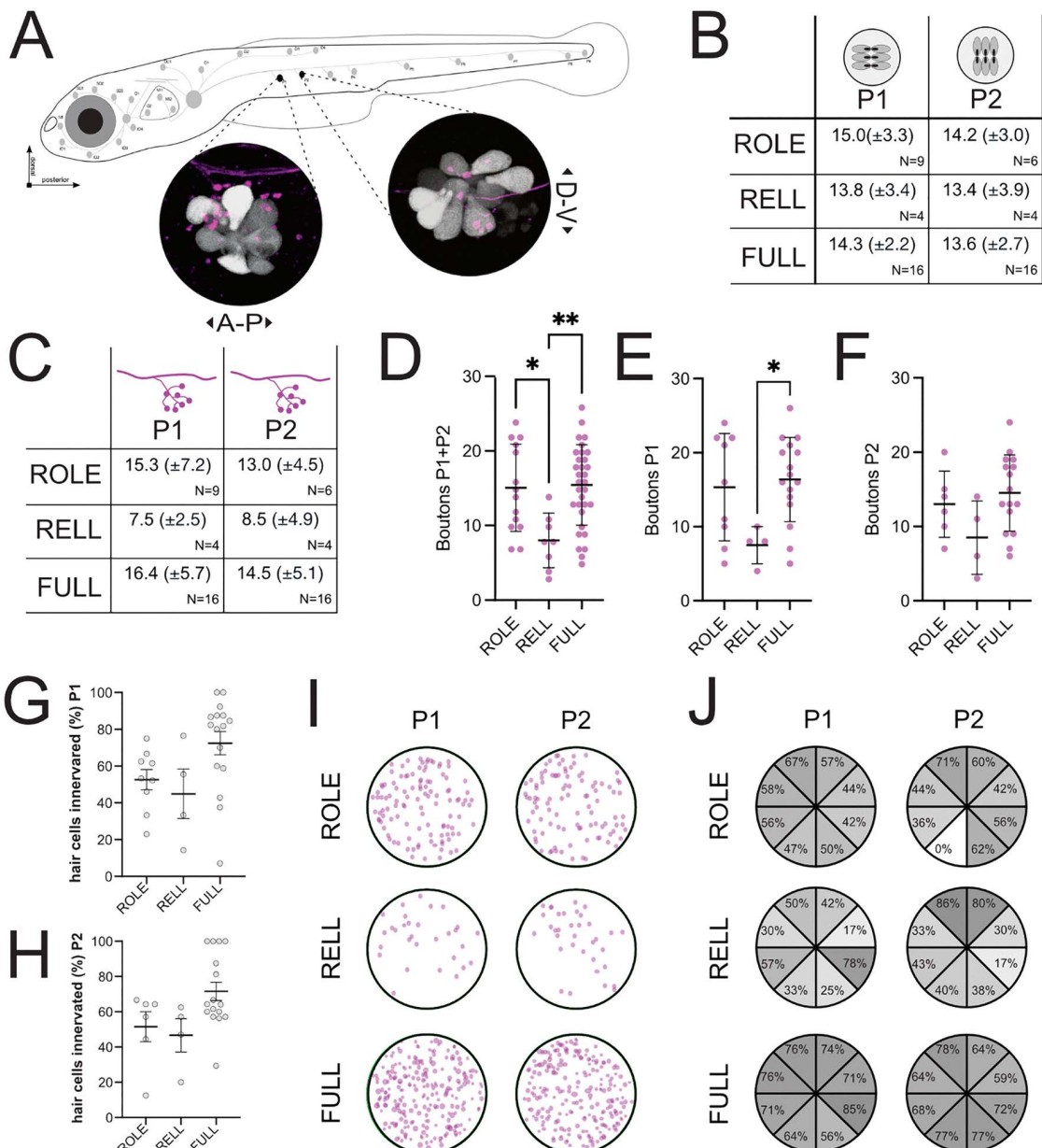

**Fig 7. Neuromast innervation by ROLE and RELL neurons. (A)** Schematic outline of the lateral line projections and neuromasts on a 5 dpf zebrafish larvae (adapted from Manuel and colleagues [11]). The P1 and P2 neuromast were selected based on their flow polarity: anterior-posterior (A-P) for P1, and dorsal-ventral (D-V) for P2. Circles contain representative confocal images of neuromasts in Tg(*myo6b*:hs:eGFFP,*dmrt3a*:GAL4,UAS:Tomato) larvae, showing individual hair cells (grey) and inhibitory efferent innervation (magenta). **(B)** Quantification of GFP-positive hair cells in the P1 and P2 neuromasts of Tg(*myo6b*:hs:eGFP) larvae (10 dpf). Values represent the average (± standard deviation) number of hair cells found among quantified neuromasts. **(C)** Number of synaptic boutons observed in neuromasts innervated by ROLE or RELL in Tg(*dmrt3a*:GAL4,UAS:Tomato) or all inhibitory efferent neurons (FULL) in Tg(*dmrt3a*:GAL4,UAS:mKate2). Data is plotted as the average (± standard deviation) number of boutons found in maximum projections of innervated neuromasts. **(D-F)** Statistical analysis of the number of boutons observed in the P1 and P2 neuromasts. **(G-H)** Quantification of the percentage of hair cells in the P1 and P2 neuromast in close proximity of synaptic boutons from ROLE, RELL, or FULL innervations. Assessment of proximity was done in 3D rendered confocal images. **(I)** Overlay of the position-maps of synaptic boutons from ROLE, RELL, or FULL innervation of neuromasts. **(J)** Overlay of the position maps of hair cells connected by ROLE, RELL, or FULL innervation of a neuromast. Percentages reflect the observations for connected hair cells in that segment of the neuromast. N = number of neuromasts for each group. Statistics: $p < 0.05$ (*), $p < 0.01$ (**).

strongly correlated to functional differences due to intrinsic properties. The similarity in molecular identity suggests that functional differences, if they exist, are likely governed by differences in upstream inputs or context-dependent network activity, rather than intrinsic cellular specialization.

Although we did not find an obvious innervation bias based towards hair cell polarity, our data hints at differences in how ROLE and RELL neurons innervate the neuromast. Both connect to the same number of hair cells, but ROLE cells appear to do this with a greater number of synaptic boutons, which suggests a potential difference in input strength. Such a difference may relate the observation that ROLE neurons project to both the ALL and PLL, while the REN and RELL project to the ALL or PLL, respectively [11]. In a speculative model, ROLE neurons could function to attenuate global neuromast sensitivity, for instance during self-induced swim activity (*i.e.*, feedforward inhibition). REN and RELL neurons would then provide more selective modulation of ALL and PLL neuromasts, potentially coming from sensory input (*i.e.*, feedback inhibition). If bouton number reflects input capacity, then a weaker RELL-mediated feedback inhibition compared to a stronger feedforward inhibition by ROLE might relate to distinct behavioural context. For example, limiting inhibitory strength during externally driven water flow could help preserve sensitivity to sudden environmental changes, such as those associated with predation. In contrast, self-generated water movements during swimming may occur more frequently or with greater intensity, potentially requiring stronger inhibitory modulation to prevent sensory overstimulation.

We should remember that this model is not derived from functional testing but represents one possible interpretation consistent with our sequencing and anatomical data. It is possible that our observations regarding innervation are caused by housing conditions. As zebrafish larvae were kept in static water, the majority of neuromast activation would be self-induced during swims. This would mean that, in our speculative model, the ROLE neuron will likely be more engaged than the RELL neuron, resulting in an increase in synaptic connections. This could be tested by keeping zebrafish larvae under housing conditions with flowing water, and assess if the number of RELL boutons increases.

Out conclusions are limited to transcriptional correlates and anatomical observations, and do not establish causal or functional differences between inhibitory efferent neuron subtypes. Future studies, utilising viral [82] or transgenic [83] tracing methods, should be able to reveal the synaptic input to inhibitory efferent neurons, while fictive locomotion setups combined with calcium imaging [84], allow for activity mapping of single REN, ROLE, and RELL neurons. Combined, these approaches can reveal if differences in input networks exist and how this could contribute to potential differences in function.

## Limitations

It is important to note, that all molecular comparisons in this study are based solely on transcriptional profiling, and no direct measurements of protein expression, synaptic physiology, or neuronal activity were performed. While scRNA sequencing provides a powerful approach for identifying molecular differences between cell types, it does not establish neuronal function. As such, the functional relevance of the observed transcriptional differences (or lack thereof) remains to be validated using approaches such as *in situ* hybridisation and electrophysiological experiments.

Here, we collected entire cells rather than isolated nuclei to maximize RNA yield and enable detection of transcripts predominantly localized to the cytosol. However, whole-cell collection carries an increased risk of co-collecting off-target RNA, such as transcripts from glia cells or synaptic boutons. While cellular mRNA is expected to greatly exceed off-target mRNA [49], off-target contamination may still show in our gene expression profiles. In addition, although the patch-seq protocol employs linear rather than exponential amplification, some degree of amplification bias cannot be excluded and may have affected the detection of low-abundance transcripts.

All cells were collected at a single developmental stage (5 dpf). At this age, the lateral line system, including its efferent innervation, is already functional [23,85], and we reasoned that transcriptional programs governing cell function would have been established. In contrast, genes relevant for fate specification and axon guidance are likely no longer expressed or detectable, which our dataset exemplifies. While earlier developmental stages might be more informative for identifying

such programmes, technical constraints limit feasibility: (1) fluorescent reporters are weakly expressed at early stages, making identifying cells difficult and (2) inhibitory efferent neurons cannot be reliably identified based on morphology prior to lateral line innervation, an event after which axon guidance cues related to projection paths are likely no longer relevant and therefore not expressed. These constraints limit our ability to capture transient developmental transcriptional states using the patch-seq approach.

Finally, the relatively small number of cells collected per inhibitory efferent neuron type (10–12 cells) should be sufficient to reveal gene-on/gene-off differences [22], but it is limited in statistical power to detect more subtle differences in gene expression. For some of the reported gene expression, we are able to achieve statistically significant differences between REN, ROLE, and RELL neurons, but overall modest differences in gene expression remain undetected. As a result, our conclusions primarily reflect genes with large expression differences (*i.e.*, gene-on/gene-off), and it remains possible that additional functionally relevant distinctions arise from more subtle transcriptional variation.

## Interactive web interface

Inspired by the interactive web-based interface by Baek and colleagues [25], we created our own: https://znn-efferent. serve.scilifelab.se/app/znn-efferent. Here users are able to explore the sequencing data and generate scaled gene expression plots for selected REN, ROLE, RELL, NP, or HB groups. In addition, a table is provided for each plot, listing the expression of each gene within individual cells or samples in the plotted group.

## Materials and methods

### Animals and husbandry

Adult zebrafish (*Danio rerio*) were kept at the Genome Engineering Zebrafish National Facility (SciLifeLab, Uppsala, Sweden) under 14 h light/10 h dark cycles and 28 °C water temperature. The housing and manipulation of zebrafish were done following the local welfare standards (ethical permits: C164/14, 14088/2019, and 5.8.18–12041/2024) and the European Union legislation (EU-Directive 201_63). The following transgenic lines were used: Tg(*dmrt3a*:GAL4) [86], Tg(UAS:Tomato) [11], Tg(UAS:mKate2) [this study], Tg(*myo6b*:hs:eGFP) [58], and Tg(HGn39D) [12]. The expression in Tg(UAS:Tomato) was mosaic, likely due to random silencing of its UAS repeats [87]. The larvae used in this study were kept in water with methylene blue and housed under constant darkness at 28 °C.

### Generation of transgenic lines

We generated Tg(UAS:mKate2) by micro-injection of plasmid: pzTol2-cmlc2:EGFP-SV40;5xUAS:mKate2-SV40 (Vector-builder Vector ID: VB220427−1422ahk). Briefly, fertilised zebrafish eggs were injected into the cell at the one-cell stage with 1 nl Tol2-plasmid mix (25 pg Tol2 mRNA, 120 pg plasmid). F0-larvae showing positive heart marker expression were raised, outcrossed to screen for germline transmission, and positive F1-larvae were kept to grow a stable transgenic line.

### Microscopy

All imaging was performed using a Leica SP8 confocal microscope (Leica Microsystems, Wetzlar, Germany). Larvae were anesthetized in 0.04% tricaine and mounted in low melting agarose (1.2%) and kept anesthetised during image acquisition. Image acquisition and processing was done using Leica's LasX software. Static confocal images were taken using a 25x water objective.

### Single cell collection

Zebrafish larvae were screened for inhibitory efferent neurons by confocal imaging at 4 dpf. The next day, pre-screened larvae (5 dpf) were anesthetized in 0.04% tricaine prepared in an extracellular recording solution (Sigma-Aldrich)

containing (in mM: 134 NaCl, 2.9 KCl, 2.1 CaCl₂, 1.2 MgCl₂, 10 HEPES, 10 glucose; pH 7.8; 290–300 mOsm/mL) [88]. Larvae were transferred to a recording chamber, and target cells were visualized using a Prime BSI Express Scientific CMOS camera (Teledyne Photometrics, USA) mounted on a 60 × water-immersion objective (LUMPlan FI, NA 0.9, Olympus) illuminated with a CoolLED pE-300white LED light source (CoolLED, UK). The skull was gently removed, and the overlaying cells covering the identified neurons were carefully cleared. Target neurons were collected with a sharp glass pipette (diameter 4–6 µm) filled with harvesting solution (in mM: 90 KCl, 10 MgCl₂; 1 U/µl RNase inhibitor). The collected cell was expelled into an RNase-free PCR tube containing 5 µl lysis buffer (iScript RT-qPCR Sample Preparation Reagent, Bio-Rad) by breaking the pipette tip in the tube and applying gentle positive pressure. Samples were centrifuged, and complete lysis was achieved through two freeze–thaw cycles.

## Antisense RNA (aRNA) amplification

We adapted a protocol by Kim and colleagues (2020) [89], where amplification of aRNA was done in two rounds using Superscript III reverse transcriptase (Life Technologies) and the MEGAscript® T7 Transcription Kit (Ambion, Life Technologies, AMB13345).

**First round of aRNA amplification.** For first-strand synthesis, 5 µl of RNA was combined with 4.9 µl RNAse-free water, 2.4 µl First-Strand buffer (5X), 1.2 µl dNTPs (10 mM), 0.45 µl DTT (100 mM), and 0.3 µl dt-T7 primer (10 ng/µl). After heating at 70 °C for 5 min and cooling on ice, 0.3 µl RNasin (2500 U), 0.45 µl Superscript III, and 1 µl RNAse-free water were added. Reactions were incubated at 42 °C for 30 min and 70 °C for 15 min.

Second-strand synthesis was performed on 9.35 µl of the first-strand reaction with 5.56 µl Second-Strand buffer (5X), 0.75 µl dNTPs (10 mM), 1 µl DNA polymerase I (10 U/µl), 0.25 µl RNase H (2 U/µl), and 8.26 µl RNAse-free water for 2 h at 16 °C. Next, 1 µl T4 DNA polymerase (5 U/µl) was added and incubated at 16 °C for 10 min.

Double-stranded DNA was purified with 52 µl Agencourt XP RNAclean beads (Beckman Coulter) and eluted in 4 µl RNAse-free water. *In vitro* transcriptase was performed by addition of 1 µl of each rNTP (10 mM), 1 µl buffer (10X), and 1 µl enzyme mix (10X) for 14 h at 37 °C. RNA was purified with 18 µl Agencourt XP RNAclean beads and eluted in 4 µl RNAse-free water.

**Second round of aRNA amplification.** First-strand synthesis (4 µl) was mixed with 1 µl random primers (0.05 µg/µl) and incubated at 70 °C for 10 min, followed by addition of 2 µl First-Strand buffer (5X), 0.5 µl dNTPs (10 mM), 0.5 µl RNasin (2500 U), 1 µl DTT (100 mM), and 1 µl Superscript III, and incubated at 25 °C for 10 min, 42 °C for 30 min, and 95 °C for 5 min.

Second-strand synthesis was performed by adding 1 µl dt-T7 oligo (10 ng/µl) to the 10 µl first-strand product, heating at 70 °C for 5 min, and adding 7.5 µl Second-Strand buffer (5X), 0.75 µl dNTPs (10 mM), 1 µl DNA polymerase (10 U/µl), and 17.25 µl RNAse-free water. Incubation proceeded for 2 h at 16 °C, followed by 1 µl T4 DNA polymerase (5 U/µl) for 10 min.

Products were purified with 70 µl RNAClean XP beads and eluted in 4 µl RNAse-free water. At this step, samples were stored at −80 °C until used in the final *in vitro* transcriptase reaction prior to single cell RNA sequencing.

The final *in vitro* transcriptase was performed by addition of 1 µl of each rNTP (10 mM), 1 µl buffer (10X), and 1 µl enzyme mix (10X) for 14 h at 37 °C. RNA was purified with 18 µl Agencourt XP RNAclean beads and eluted 50 µl RNAse-free water. Final concentration was measured using a NanoDrop Spectrophotometer (ND-1000, NanoDrop Technologies) and samples were transferred to the 96-well sequencing plate.

## Single cell sequencing

Sequencing (2022-11-10) was performed by the SNP&SEQ Technology Platform (Uppsala, Sweden). Libraries were prepared from 100ng total RNA using the TruSeq stranded mRNA library preparation kit (cat# 20020595, Illumina Inc.) including polyA selection. Unique dual indexes (cat# 20022371, Illumina Inc.) were used. The library preparation was performed

according to the manufacturers' protocol (#1000000040498). Sequencing details: paired-end 150 bp read length, Nova-Seq 6000 system, S4 flowcell and v1.5 sequencing chemistry. A sequencing library for the phage PhiX was included as 1% spike-in in the sequencing run.

## Sequence data analysis

**Reference genome.** We downloaded the zebrafish reference genome (version 11) from Ensembl as well as the annotations (release 109; download data: 2023-03-23). The analysis pipeline used was the nf-core framework [90], available at: https://doi.org/10.5281/zenodo.1400710 and https://github.com/nf-core/.

**Post mapping QC.** QC of the reads was done with FastQC (0.11.9), mapping to the genome was done with STAR (2.7.9a), post-mapping QC was done with Preseq (3.1.1), RSeQC (3.0.1), and QualiMap (2.2.2-dev). Expression values were generated by Salmon (1.4.0) [91] in the nf-core/rnaseq (version 3.10.1) pipeline and a summary of the analysis was performed by MultiQC (3.9.5). Then we further checked the quality of the data using Seurat (v4) to screen for proportion of mitochondrial genes, rRNA, and ribosomal proteins. Also, we examined cell-cycle status of the cells using gene markers in Seurat package. [92]. Analysis was done with R (4.2.2) in platform: x86_64-apple-darwin17.0 (64-bit), running under: macOS Big Sur 10.16.

**Filtering.** For downstream analysis, we only considered cells/samples with at least 100 detected genes and analysed genes that had expression in at least 3 cells.

**Differential expression.** We performed differential expression analysis by using Wilcoxon Rank Sum test, MAST, as well as DESeq2 methods. To select differentially expressed genes (DEGs), we used following cut-offs: logFC > |1|, DR(p_adj_val) <= 0.05, and minimum 20% of cells in either group should have expression.

## Whole-mount immunohistochemistry

Embryos of Tg(dmrt3a:GAL4,UAS:mKate2) were housed at 28 °C in embryo water. At 3dpf, larvae were euthanised with 0.2% tricaine and fixed in 4% paraformaldehyde (in PBS) for 1 hour at room temperature and washed 3x10 min in PBS and 3x10 min in 0.25% PBS-T (pH 7.3, TritonX-100). Next, larvae were kept in 150 mM TrisHCl (pH 8.0) for 5 min at RT and 20 min at 65 °C. Samples were washed 3x10 min in 0.25% PBS-T. Zebrafish were permeabilised on ice for 45 min while kept in PBS (0.1% Trypsin 10X, 2.5 mM EDTA). Following 3x10 min washes in PBS-T, larvae were placed in blocking buffer (PBS-T with 2% FCS, 2% DMSO). After blocking, embryos were incubated overnight with the primary antibody (1:100 rabbit gad1b polyclonal antibody (Invitrogen PA5143365)) in blocking buffer at 4 °C. On the second day, larvae were washed 5x20 min in PBS-T and incubated overnight with a secondary antibody (1:1000, FITC donkey anti-rabbit) in blocking buffer at 4 °C. On the third day, larvae were washed 5x20 min with PBS-T and imaged in a Leica SP8 confocal microscope. All incubation steps at RT and 4 °C were performed under gentle agitation and in an incubation box (patent number 2151279−3).

## Neuromast innervation

To prevent pigmentation, 1-Phenyl-2-thiourea (PTU, 0.003% final concentration) was added at 24 h post fertilization (hpf). Zebrafish larvae were then screened for ROLE or RELL efferent neurons by confocal imaging of the rhombomere region at 5 dpf. Subsequently, selected larvae were placed individually in a well of a 6-well plate, supplied with fresh embryo water (0.003% PTU), and fed with 1–2 drops of rotifer culture and left till day 10. At 10 dpf, the pre-screened larvae were once more imaged, but this time laterally to image efferent innervation of P1 and P2 neuromasts.

## Analysis of neuromast innervation

All images were orientated so that the rostral to caudal axis (left to right) and dorsal to ventral axis (top to bottom) were the same for each neuromast analysed. With LasX 3D rendering, the kinocilium of hair cells were aligned toward the screen. The maximum projection of this orientation and the slice of base synaptic plane was exported to use as a template

to identify individual hair cells and their innervation by inhibitory neurons. Quantification of the number of synapse boutons was done manually, using the previously exported maximum projections. Annotation of the number of connected hair cells in each neuromast was done manually, by going through each Z-stack layer in the LasX 3D render.

## Supporting information

**S1 File. Supporting data.** Supporting data for nReads and nGenes for each sample and for P1 and P2 neuromast innervation (hair cell and synapse bouton counts) for ROLE and RELL neurons.
(XLSX)

## Acknowledgments

We thank the Zebrafish Core Facility (CIV, Uppsala, Sweden) for fish husbandry. Sequencing was performed by the SNP&SEQ Technology Platform in Uppsala, which is part of the National Genomics Infrastructure (NGI) Sweden and Science for Life Laboratory (SciLifeLab). Computations were performed on resources provided by SNIC through Uppsala Multidisciplinary Center for Advanced Computational Science (UPPMAX) under Project 2022/23–528 (Computation) & 2022/22–1037 (Storage). This work was further supported by the National Bioinformatics Infrastructure Sweden (NBIS) at SciLifeLab.

## Author contributions

**Conceptualization:** Remy Manuel, Henrik Boije.

**Data curation:** Remy Manuel, Melek Umay Tuz-Sasik.

**Formal analysis:** Remy Manuel, Melek Umay Tuz-Sasik.

**Funding acquisition:** Remy Manuel, Henrik Boije.

**Investigation:** Remy Manuel, Aikeremu Ahemaiti, Melek Umay Tuz-Sasik.

**Methodology:** Remy Manuel, Aikeremu Ahemaiti, Melek Umay Tuz-Sasik.

**Project administration:** Remy Manuel, Henrik Boije.

**Resources:** Remy Manuel, Henrik Boije.

**Supervision:** Remy Manuel, Henrik Boije.

**Validation:** Remy Manuel, Aikeremu Ahemaiti, Melek Umay Tuz-Sasik, Henrik Boije.

**Visualization:** Remy Manuel, Melek Umay Tuz-Sasik.

**Writing – original draft:** Remy Manuel.

**Writing – review & editing:** Aikeremu Ahemaiti, Melek Umay Tuz-Sasik, Henrik Boije.

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
