## [Decision Letter · Decision Letter 0]

12 Nov 2025

Dear Dr. Manuel,

We look forward to receiving your revised manuscript.

Kind regards,

Annie Angers, Ph.D.

Academic Editor

PLOS ONE

Journal Requirements:

2. Please note that PLOS One has specific guidelines on code sharing for submissions in which author-generated code underpins the findings in the manuscript. In these cases, we expect all author-generated code to be made available without restrictions upon publication of the work.

Please review our guidelines at https://journals.plos.org/plosone/s/materials-and-software-sharing#loc-sharing-code and ensure that your code is shared in a way that follows best practice and facilitates reproducibility and reuse.

“Financial support came from: the Kjell and Marta Beijers Foundation; the Jeanssons Foundation; the Carl Tryggers Foundation; the Swedish Brain Foundation; the Swedish Research Council; the Magnus Bergvalls Foundation, the Royal Swedish Academy of Sciences; the Ake Wibergs Foundation; Olle Engkvist Stiftelse; the Ragnar Söderberg Foundation, and the Swedish Foundation for Strategic Research. The SNP&SEQ Platform is also supported by the Swedish Research Council and the Knut and Alice Wallenberg Foundation.”

4. Please note that funding information should not appear in the Acknowledgments section or other areas of your manuscript. We will only publish funding information present in the Funding Statement section of the online submission form. Please remove any funding-related text from the manuscript.

5. We note that you have indicated that there are restrictions to data sharing for this study. PLOS only allows data to be available upon request if there are legal or ethical restrictions on sharing data publicly. For more information on unacceptable data access restrictions, please see http://journals.plos.org/plosone/s/data-availability#loc-unacceptable-data-access-restrictions.

6. Please note that your Data Availability Statement is currently missing the direct link to access each database. If your manuscript is accepted for publication, you will be asked to provide these details on a very short timeline. We therefore suggest that you provide this information now, though we will not hold up the peer review process if you are unable.

7. We note that you have included the phrase “data not shown” in your manuscript. Unfortunately, this does not meet our data sharing requirements. PLOS does not permit references to inaccessible data. We require that authors provide all relevant data within the paper, Supporting Information files, or in an acceptable, public repository. Please add a citation to support this phrase or upload the data that corresponds with these findings to a stable repository (such as Figshare or Dryad) and provide and URLs, DOIs, or accession numbers that may be used to access these data. Or, if the data are not a core part of the research being presented in your study, we ask that you remove the phrase that refers to these data.

**Additional Editor Comments:**

Thank you for submitting this very interesting study. All reviewers praise the work done and consider your study highly valuable. However, they raise some concerns that need to be addressed before publication. Specifically, it is emphasized that the transcriptomics findings should be validated in vivo, and that it should be demonstrated whether the cell count, read depth, reads per cell, and gene representation are sufficient to provide the statistical power needed to detect differences in these populations, if such differences exist.

Reviewers' comments:

Reviewer's Responses to Questions

**Comments to the Author**

1. Is the manuscript technically sound, and do the data support the conclusions?

Reviewer #1: Partly

Reviewer #2: Partly

Reviewer #3: No

2. Has the statistical analysis been performed appropriately and rigorously?

Reviewer #1: Yes

Reviewer #2: Yes

Reviewer #3: No

3. Have the authors made all data underlying the findings in their manuscript fully available?

Reviewer #1: Yes

Reviewer #2: Yes

Reviewer #3: No

4. Is the manuscript presented in an intelligible fashion and written in standard English?

Reviewer #1: Yes

Reviewer #2: Yes

Reviewer #3: Yes

Reviewer #1: General Comments

This manuscript by Manuel et al. presents a valuable investigation into the functional differentiation of three inhibitory efferent neuron subtypes (REN, ROLE, and RELL) in the zebrafish lateral line. The authors employ a dual strategy, combining single-cell transcriptomics with an anatomical analysis of neuromast innervation patterns. The study is well-designed, addressing a clear and important question in sensory neurobiology. The finding that these morphologically distinct neuron subtypes are remarkably similar at both the molecular and anatomical levels is a significant contribution, highlighting that functional specificity may arise from upstream inputs rather than intrinsic cellular properties.

The technically challenging approach of manually collecting visually-identified single neurons is commendable, as it ensures exceptional sample purity for sequencing. The manuscript is well-written, and the conclusions are generally drawn with appropriate caution. The work is potentially suitable for publication in PLOS ONE, provided the major concerns outlined below are addressed.

Major Comments

Requirement for In Vivo Validation of Key Genes: The primary weakness of this study is the lack of validation for its novel transcriptomic findings. It is essential to perform in vivo validation via methods such as HCR-FISH for the most critical discoveries, including the expression of GABA-related genes (gad1b, gad2), as well as key differentially expressed genes (DEGs) and membrane potential-related genes that suggest subtle differences between subtypes. This validation is critical for substantiating the study's conclusions. If this is not feasible, the authors must clearly state that these findings are hypotheses generated from the transcriptomic data, not established facts.

Minor Comments

aRNA Amplification Bias: While necessary for such low-input samples, the two rounds of aRNA amplification are known to introduce potential bias. A brief acknowledgment of this limitation in the Methods or Discussion would enhance the study's transparency.

Interpretation of Anatomical Differences: The authors conclude that the neuron subtypes are molecularly similar, yet they report a significant anatomical difference (fewer synaptic boutons for RELL neurons). The manuscript would be strengthened if the authors added a discussion on how this subtle structural difference might arise despite the transcriptomic similarity and what its potential functional implications might be.

Reviewer #2: This study aims to investigate the differences of the three Inhibitory efferent neurons, REN, ROLE, and RELLby single-cell RNA sequencing in 5 dpf zebrafish larvae. The work is conceptually interesting and presents a potentially valuable hypothesis. However, in its current form, the manuscript is a little simple and lacks the methodological rigor for publication. The study is primarily descriptive of sequencing results, and the results are not adequately supported by quantitative or histological stains. My comments are as follows:

1. In the abstract, the author emphasizes that no strong molecular evidence for functional differences is found between the three types of inhibitory efferent neurons, and genes related to membrane potentials were equally expressed across REN, ROLE, and RELL cells. I think this description is inaccurate. The ScRNA Seq results showed that there are significantly different expressed genes in Figure 6.

2. Different maturity levels may lead to different phenotypes. In this study, the authors only chose the 5 dpf time point for RNA sequencing. Why? The author should add the RNA sequencing data at a late stage.

3. This study uses whole-cell scRNA-seq; however, for cells with long protrusions, such as neurons, which are prone to the loss of cytoplasmic mRNA, snRNA-seq is often more robust. Why was snRNA-seq not chosen? Could the differences between these two sequencing methods affect the conclusions?

4. In this study, the authors investigate the differences of the three inhibitory efferent neurons, REN, ROLE, and RELL, only by ScRNA-Seq. It’s not enough. The authors must add the Immunohistochemistry or in situ hybridization experiments to further confirm the results of scRNA-Seq in Figure 2- 6.

5. In Figure 6, the authors should add the Electrophysiological experiment to check the scRNA-seq results.

6. In Figure 7 (The author mistakenly wrote it as Figure 6), the authors explored whether the efferent neurons displayed unique anatomical innervation of neuromasts by confocal imaging. However, only diagrams are displayed in this study. The representative pictures should be added.

7. The cell numbers should be added in Figure 3A.

8. The full term should be provided when an abbreviation is used for the first time.

Reviewer #3: This manuscript uses single cell RNA sequencing to characterize gene expression and possible gene expression differences in the efferent inhibitory neurons REN, ROLE, and RELL that innervate the zebrafish neuromast sensory organs in the lateral line. Understanding how these neurons differentiate and differentially innervate the neuromasts is an important goal.

The authors use a single-cell extraction technique to physically remove the individual neurons identified by shape from the hindbrain. Using a single cell RNA seq technique developed for patch clamp extraction of cytoplasm from neurons, they sequenced the RNAs of each cell and characterized gene expression differences. Upon PCA clustering, the different neuron types displayed no significant clustering, suggesting that they largely had the same gene expression profiles. The authors conclusions are that these neurons were identified as inhibitory cholinergic by neurotransmitter phenotype, but no other significant gene expression differences existed to explain their different innervation patterns and morphologies.

Very few cells in each class were analyzed, only 4-16 cells for each class. Furthermore, there is no indication of read depth, number of reads for each cell, and number of genes in the genome represented by the technique. This small cell number and potentially small read and gene representation number cause concern for the statistical power to detect changes in gene expression, especially if the differences are subtle which is to be expected of similar neuron classes. There is no indication of statistical power or resolution to detect expression differences based upon these low cell numbers and possibly low read counts. Lack of statistical power could explain the lack of gene expression differences detected between the classes. This undermines the statistical power and rigor of the conclusions about gene expression differences. If there is a published example of low cell numbers like this detecting significant differences between neurons, it should be discussed. Without the data regarding read counts and statistical power, these results do not support the conclusions of the authors that no significant differences exist.

Other comments:

The authors say they use the “top 2000 genes”, but representation of 100 genes was the cut-off. Please explain.

Some DEGs are identified, but the authors say they are not important. This needs to be explained. Was a GO term analysis on these genes conducted on these genes? Why do the authors dismiss them?

Using percentages as indications of number of cells expressing a gene is problematic, given the small cell number. Maybe percentage ranges could be used instead.

The patch clamp RNA seq method is a non-standard way to do RNA seq, basically first making a full-length cDNA, and then conducting in vitro transcription to produce RNAs that are then sequenced. I understand that the technique prevents bias of PCR amplification, but use of this technique needs to be justified in analysis of differential gene expression in scRNA seq.

The characterization of gene classes in these neurons is fine, but is still plagued by low cell numbers and read counts.

A table showing cell number, counts for each cell, and gene representation for each cell would help. along with a discussion of statistical power to detect gene expression differences with these low cell numbers.

.

Reviewer #1: No

Reviewer #2: No

Reviewer #3: No

---

## [Author Response · Author response to Decision Letter 1]

3 Mar 2026

** COPIED FROM UPLOADED DOCUMENT **

Reviewer #1: General Comments

This manuscript by Manuel et al. presents a valuable investigation into the functional differentiation of three inhibitory efferent neuron subtypes (REN, ROLE, and RELL) in the zebrafish lateral line. The authors employ a dual strategy, combining single-cell transcriptomics with an anatomical analysis of neuromast innervation patterns. The study is well-designed, addressing a clear and important question in sensory neurobiology. The finding that these morphologically distinct neuron subtypes are remarkably similar at both the molecular and anatomical levels is a significant contribution, highlighting that functional specificity may arise from upstream inputs rather than intrinsic cellular properties.

The technically challenging approach of manually collecting visually-identified single neurons is commendable, as it ensures exceptional sample purity for sequencing. The manuscript is well-written, and the conclusions are generally drawn with appropriate caution. The work is potentially suitable for publication in PLOS ONE, provided the major concerns outlined below are addressed.

Major Comments

Requirement for In Vivo Validation of Key Genes: The primary weakness of this study is the lack of validation for its novel transcriptomic findings. It is essential to perform in vivo validation via methods such as HCR-FISH for the most critical discoveries, including the expression of GABA-related genes (gad1b, gad2), as well as key differentially expressed genes (DEGs) and membrane potential-related genes that suggest subtle differences between subtypes. This validation is critical for substantiating the study's conclusions. If this is not feasible, the authors must clearly state that these findings are hypotheses generated from the transcriptomic data, not established facts.

Reply: We agree with the reviewer, that verifying significant observations made in our sequencing dataset should ideally be followed up with in situ hybridization studies. Had we found convincing differential expression supporting our idea of functional differences at a cellular level, or differences in fate influencing transcription factors or axon guidance receptors, we would have pursued follow-up experiments to confirm these findings. However, as this is not the case, we decided to share our sequencing results as they are. That said, we did now use antibodies in an attempt to show Gad1b within the efferent neurons, as we felt this was one of the more crucial observations we made. These results have been included in the revised version of the manuscript (Figure 4). We have now stipulated that our conclusions are drawn from sequencing data only, adjusted the text throughout the manuscript to reflect this, and included a newly added “Limitations Sections”.

Minor Comments

aRNA Amplification Bias: While necessary for such low-input samples, the two rounds of aRNA amplification are known to introduce potential bias. A brief acknowledgment of this limitation in the Methods or Discussion would enhance the study's transparency.

Reply: The method we used should limit the introduction of Amplification Bias, as it employs linear amplifications, rather than exponential. We have now touched upon this in a newly added Limitations section. In addition, we included some additional information regarding number of reads and number of genes in each sample, as well as the correlation between nReads and nGenes (Figure 2).

Interpretation of Anatomical Differences: The authors conclude that the neuron subtypes are molecularly similar, yet they report a significant anatomical difference (fewer synaptic boutons for RELL neurons). The manuscript would be strengthened if the authors added a discussion on how this subtle structural difference might arise despite the transcriptomic similarity and what its potential functional implications might be.

Reply: We have now expanded on our previous text where we discuss why there would be innervation differences, and included text providing an explanation for how they came to be.

Reviewer #2: This study aims to investigate the differences of the three Inhibitory efferent neurons, REN, ROLE, and RELLby single-cell RNA sequencing in 5 dpf zebrafish larvae. The work is conceptually interesting and presents a potentially valuable hypothesis. However, in its current form, the manuscript is a little simple and lacks the methodological rigor for publication. The study is primarily descriptive of sequencing results, and the results are not adequately supported by quantitative or histological stains. My comments are as follows:

1. In the abstract, the author emphasizes that no strong molecular evidence for functional differences is found between the three types of inhibitory efferent neurons, and genes related to membrane potentials were equally expressed across REN, ROLE, and RELL cells. I think this description is inaccurate. The ScRNA Seq results showed that there are significantly different expressed genes in Figure 6.

Reply: The Reviewer is correct and we have adjusted our phrasing in the Abstract regarding gene expression of genes shown in figure 6. However, we maintain our overall conclusion, that the inhibitory efferent neurons show no strong differential gene expression which indicates relevant functional, fate influencing, or morphological differences.

2. Different maturity levels may lead to different phenotypes. In this study, the authors only chose the 5 dpf time point for RNA sequencing. Why? The author should add the RNA sequencing data at a late stage.

Reply: Our aim with this study, was to identify genes that could point towards functional, fate influencing and morphological differences. At 5 dpf, the lateral line system of zebrafish appears functional, and although still expanding, it contains all components. We reasoned that, at this stage, differences between neurons would be established and we would be able to reveal differential expression. That said, we do agree with the Reviewer that larvae are still developing and there is a chance the expression profiles between neurons become more unique and different as the fish matures. We have acknowledged this limitation in a newly added Limitations Section.

3. This study uses whole-cell scRNA-seq; however, for cells with long protrusions, such as neurons, which are prone to the loss of cytoplasmic mRNA, snRNA-seq is often more robust. Why was snRNA-seq not chosen? Could the differences between these two sequencing methods affect the conclusions?

Reply: Collecting nuclear RNA circumvents many issues when dissociating neurons, such as stressing the cells and ending up with neurons free from glia. However, in our approach we did not dissociate, but instead directly collected individual neurons from intact brain tissue. Because of this, we argued that many of the issues encountered during tissue dissociation were avoided. Still, assessing nuclear RNA may have been less prone to contamination by transcripts from neighbouring cells or synapses from other neurons. However, nuclear RNA generally provides fewer transcripts and we would lose transcripts predominantly found in the cytosol. We figured this would cause problems downstream, as we already have to perform an amplification step to generate sufficed RNA for single cell sequencing. It is hard to determine if nuclear RNA would have resulted in different gene expression profile, as it would depend on the abundance of specific transcripts in the nucleus versus the cytosol. The samples would likely be less prone to contamination, but we would also have fewer detected genes that are cell specific. We touch upon this in the newly added Limitations section.

4. In this study, the authors investigate the differences of the three inhibitory efferent neurons, REN, ROLE, and RELL, only by ScRNA-Seq. It’s not enough. The authors must add the Immunohistochemistry or in situ hybridization experiments to further confirm the results of scRNA-Seq in Figure 2- 6.

5. In Figure 6, the authors should add the Electrophysiological experiment to check the scRNA-seq results.

Reply: We agree with the reviewer, that verifying significant observations made in our sequencing dataset, would ideally be followed up with in situ hybridization studies (and functional studies). Had we uncovered major differential expression supporting our idea of functional differences at a cellular level, we would have pursued follow-up experiments to confirm. But as we did not find genes that could explain differences in fate assignment, axon guidance, or major functional differences, we decided to share our sequencing results as they are. That said, we did now use antibodies in an attempt to show Gad1b within the efferent neurons, as we felt this was one of the more crucial observations we made. These results have been included in the revised version of the manuscript (Figure 4). We have now stipulated that our conclusions are drawn from sequencing data only, adjusted the text throughout the manuscript to reflect this, and included a newly added “Limitations Sections”.

6. In Figure 7 (The author mistakenly wrote it as Figure 6), the authors explored whether the efferent neurons displayed unique anatomical innervation of neuromasts by confocal imaging. However, only diagrams are displayed in this study. The representative pictures should be added.

Reply: We have corrected the figure number and included a representative picture of axon innervation of neuromasts.

7. The cell numbers should be added in Figure 3A.

Reply: We have added the number of cells/samples for each group in figure 3A

8. The full term should be provided when an abbreviation is used for the first time.

Reply: We went through the Manuscript and added full terms when using abbreviations for the first time. The only exception to this is where we use the name for genes, as we felt the loss of readability was worse than not providing the full gene name in every instance.

Reviewer #3: This manuscript uses single cell RNA sequencing to characterize gene expression and possible gene expression differences in the efferent inhibitory neurons REN, ROLE, and RELL that innervate the zebrafish neuromast sensory organs in the lateral line. Understanding how these neurons differentiate and differentially innervate the neuromasts is an important goal.

The authors use a single-cell extraction technique to physically remove the individual neurons identified by shape from the hindbrain. Using a single cell RNA seq technique developed for patch clamp extraction of cytoplasm from neurons, they sequenced the RNAs of each cell and characterized gene expression differences. Upon PCA clustering, the different neuron types displayed no significant clustering, suggesting that they largely had the same gene expression profiles. The authors conclusions are that these neurons were identified as inhibitory cholinergic by neurotransmitter phenotype, but no other significant gene expression differences existed to explain their different innervation patterns and morphologies.

Very few cells in each class were analyzed, only 4-16 cells for each class. Furthermore, there is no indication of read depth, number of reads for each cell, and number of genes in the genome represented by the technique.

Reply: When we designed the experiment, we believed our sample size to be sufficient to reveal major gene expression differences that would indicate functional differences between REN, ROLE and RELL neurons. The most relevant groups (i.e. those containing single cell samples) contain 11-16 samples. The 4 samples in the HB (hindbrain) group are samples consisting of thousands of neurons, and served as a reference for genes expressed in the hindbrain as a whole. In regards to the read depth and reads per cell, we now include a graph in the manuscript (Figure 2).

This small cell number and potentially small read and gene representation number cause concern for the statistical power to detect changes in gene expression, especially if the differences are subtle which is to be expected of similar neuron classes.

There is no indication of statistical power or resolution to detect expression differences based upon these low cell numbers and possibly low read counts. Lack of statistical power could explain the lack of gene expression differences detected between the classes. This undermines the statistical power and rigor of the conclusions about gene expression differences. If there is a published example of low cell numbers like this detecting significant differences between neurons, it should be discussed. Without the data regarding read counts and statistical power, these results do not support the conclusions of the authors that no significant differences exist.

Reply: There are studies that use similar number of neurons to identify differences in gene expression. There is a paper listing several studies with a low number of cells used for patch-seq experiments (10.1523/JNEUROSCI.1653-20.2020). For instance, Oláh and colleagues (https://doi.org/10.7554/eLife.58515) used 8 neurons per group to reveal significant differences in molecular markers. It should be noted, that the neurons of these two groups already showed differences in electrophysiological properties, making it more likely differences in expression would exist. Regardless, if similar functional differences on an electrophysiological level would exist between our neurons, 11-16 samples per group should have been sufficient to reveal these.

Other comments:

The authors say they use the “top 2000 genes”, but representation of 100 genes was the cut-off. Please explain.

Reply: First, we checked if each sample contained at least 100 genes that were also identified in at least 2 additional samples. If a sample did not show at least 100 genes that were also expressed in other samples, we would exclude these samples from our down-stream analysis. We did not find such samples (we included a comment on this in the revised version). Next, we used the top 2000 variable genes in order to cluster samples in a PCA. Variable genes were identified based on how their expression changes between samples. Variation within a single sample only shows the gene expression distribution and does not provide meaningful statistical information for PCA or other downstream analyses. We included a comment on what we mean by variable genes.

Some DEGs are identified, but the authors say they are not important. This needs to be explained. Was a GO term analysis on these genes conducted on these genes? Why do the authors dismiss them?

Reply: The DEG identified do not correllate to cell fate assignment, axon guidance, or neurophenotype. We now have performed a GO analysis using the ShinyGP 0.85.1 (https://bioinformatics.sdstate.edu/go/) application. Below are the results:

• 3B: None

• 3C: 14 processes including: tripeptide transport, oligopeptide transport, maturation of lsu-rRNA and 5.8s rRNA (nGenes 1 each).

• 3D: 15 processes including: myelination periphery axons, schwann cell development, lateral line glia cell development, oligodendrocyte development (nGenes 2 each). 1 process involves Membrane organization (nGenes 4).

• 3E None

• 3F: substrate adhesion-dependent cell spreading (nGenes 4), Cell substrate adhesion (nGenes 4)

We have included these findings in the figure legend.

Using percentages as indications of number of cells expressing a gene is problematic, given the small cell number. Maybe percentage ranges could be used instead.

Reply: We agree that using percentages is not ideal, however, we are not sure how reporting a percentage range would improve our data presentation. The size of the expression dots in the figures are scaled to their exact percentage of cells within their group population. The legend shows the size of the dots for the listed percentage. If readers need to know the exact number, this can be obtained via the interactive web interface, which will also provide the expression level for each individual sample.

The patch clamp RNA seq method is a non-standard way to do RNA seq, basically first making a full-length cDNA, and then conducting in vitro transcription to produce RNAs that are then sequenced. I understand that t

---

## [Decision Letter · Decision Letter 1]

17 Mar 2026

SINGLE-CELL ANALYSIS OF INHIBITORY EFFERENT NEURONS OF THE ZEBRAFISH LATERAL LINE

PONE-D-25-53779R1

Dear Dr. Manuel,

We’re pleased to inform you that your manuscript has been judged scientifically suitable for publication and will be formally accepted for publication once it meets all outstanding technical requirements.

Kind regards,

Annie Angers, Ph.D.

Academic Editor

PLOS One

Additional Editor Comments (optional):

Reviewers' comments:

Reviewer's Responses to Questions

**Comments to the Author**

Reviewer #1: All comments have been addressed

Reviewer #2: All comments have been addressed

Reviewer #3: All comments have been addressed

2. Is the manuscript technically sound, and do the data support the conclusions?

Reviewer #1: Yes

Reviewer #2: Partly

Reviewer #3: Yes

3. Has the statistical analysis been performed appropriately and rigorously?

Reviewer #1: Yes

Reviewer #2: Yes

Reviewer #3: Yes

4. Have the authors made all data underlying the findings in their manuscript fully available?

Reviewer #1: Yes

Reviewer #2: Yes

Reviewer #3: Yes

5. Is the manuscript presented in an intelligible fashion and written in standard English?

Reviewer #1: Yes

Reviewer #2: Yes

Reviewer #3: Yes

Reviewer #1: (No Response)

Reviewer #2: Most of the authors’ responses are satisfactory to me. Although the authors have provided some explanation for my comments 4 and 5, I still believe that, to further strengthen the persuasiveness of the manuscript, additional histological or functional validation is needed.

Reviewer #3: (No Response)

.

Reviewer #1: No

Reviewer #2: No

Reviewer #3: No

---

## [Editor Report · Acceptance letter]

PONE-D-25-53779R1

PLOS One

Dear Dr. Manuel,

I'm pleased to inform you that your manuscript has been deemed suitable for publication in PLOS One. Congratulations! Your manuscript is now being handed over to our production team.

Kind regards,

on behalf of

Dr. Annie Angers

Academic Editor

PLOS One